# Mitigation of Airborne PRRSV Transmission with UV Light Treatment: Proof-of-Concept

Peiyang Li [1], Jacek A. Koziel [1,*], Jeffrey J. Zimmerman [2], Jianqiang Zhang [2], Ting-Yu Cheng [2], Wannarat Yim-Im [2], William S. Jenks [3], Myeongseong Lee [1], Baitong Chen [1] and Steven J. Hoff [1]

[1] Department of Agricultural and Biosystems Engineering, Iowa State University, Ames, IA 50011, USA; peiyangl@iastate.edu (P.L.); leefame@iastate.edu (M.L.); baitongc@iastate.edu (B.C.); hoffer@iastate.edu (S.J.H.)

[2] Department of Veterinary Diagnostic and Production Animal Medicine, Iowa State University, Ames, IA 50011, USA; jjzimm@iastate.edu (J.J.Z.); jqzhang@iastate.edu (J.Z.); cheng.1784@osu.edu (T.-Y.C.); wyimim@iastate.edu (W.Y.-I.)

[3] Department of Chemistry, Iowa State University, Ames, IA 50011, USA; wsjenks@iastate.edu

* Correspondence: koziel@iastate.edu; Tel.: +1-515-294-4206

**Abstract:** Proper treatment of infectious air could potentially mitigate the spread of airborne viruses such as porcine reproductive and respiratory syndrome virus (PRRSV). The objective of this research is to test the effectiveness of ultraviolet (UV) in inactivating aerosolized PRRSV, specifically, four UV lamps, UV-A (365 nm, both fluorescent and LED-based), "excimer" UV-C (222 nm), and germicidal UV-C (254 nm), were tested. The two UV-C lamps effectively irradiated fast-moving PRRSV aerosols with short treatment times (<2 s). One-stage and two-stage UV inactivation models estimated the UV doses needed for target percentage (%) reductions on PRRSV titer. UV-C (254 nm) dose needed for 3-log (99.9%) reduction was 19.43 and 2.44 mJ/cm$^2$, respectively, based on one-stage and two-stage models. UV-C (222 nm) doses needed for a 3-log reduction 2.81 and 1.04 mJ/cm$^2$, based on one-stage and two-stage models, respectively. However, the cost of 222 nm excimer lamps is still economically prohibitive for scaling-up trials. The UV-A (365 nm) lamps could not reduce PRRSV titers for tested doses up to 4.11 mJ/cm$^2$. Pilot-scale or farm-scale testing of UV-C on PRRSV aerosols simulating barn ventilation rates are recommended based on its effectiveness and reasonable costs comparable to HEPA filtration.

**Keywords:** air purification; animal production; porcine reproductive; respiratory syndrome; livestock health; livestock biosecurity; swine diseases; ultraviolet light



## 1. Introduction

Since its initial documentation in the late 1980s [1,2], porcine reproductive and respiratory syndrome (PRRS) has been one of the most impactful diseases affecting the United States of America (USA)'s swine industry. The disease has two overlapping clinical symptoms, reproductive failure or impairment, as well as respiratory diseases. The annual cost of PRRS disease to the producers in the USA was estimated to be USD 560 million estimated in 2005 [3], USD 664 million in 2012 [4], and USD 580 million in 2016 [5]. The cost per pig ranged from USD 6.25 to 15.25 in the market [4,6]. The battle against PRRS is still among the top priorities for producers and veterinarians.

Porcine reproductive and respiratory syndrome virus (PRRSV) can be transmitted via indirect contact (such as aerosol and fomites) or direct contact, but it likewise is found in aerosols generated by infected pigs and can reach susceptible pigs meters or perhaps kilometers away [7–10]. Research suggests that infectious PRRSV aerosols may travel up to ~9 km [11,12]. Given its infectivity and airborne survivability, proper treatment or decontamination of PRRSV aerosols could effectively reduce transmission.

Mitigation against PRRS has been investigated at both laboratory and farm scales, including but not limited to air filtration, air ionization [13], non-thermal plasma (NTP) reactor [14], UV treatment [15], vaccination [16], and depopulation [17].

In terms of air filtration, high-efficiency particulate air (HEPA) filters have been used by selected few swine operations to treat incoming ventilation air and prevent barns from infectious airborne viruses. These filters were proven to effectively prevent the transmission of PRRSV [18,19]. However, maintenance of these filters is labor-intensive and requires expensive addition to barn space. Thus, very few facilities in the USA can afford mechanical filtration of incoming air. A more affordable way of defending swine barns from incoming infectious air is needed for the industry.

Ultraviolet (UV) light, especially ultraviolet-C (UV-C, ranging from 200–280 nm) [20,21], has been proven to disinfect or inactivate pathogens. In theory, UV can potentially reduce operational and capital costs by eliminating the implementation of high-grade filters. To date, only one paper has been published on the use of UV for aerosolized PPRSV inactivation [22]. At present, Cutler et al. (2012) [22] remains the benchmark study on UV-C treatment of aerosolized PRRSV. UV-C (254 nm) inactivation and parameters under static conditions (i.e., no aerosol generation) were reported by Cutler et al. (2011) [23] for both one-stage and two-stage models. Under dynamic conditions (i.e., irradiation on PRRSV aerosols), only a one-stage model was reported by Cutler et al. (2012) [22]. The estimations on UV-C (254 nm) doses needed to inactivate practical PRRSV levels (e.g., 90, 99, 99.9%) remain very limited, i.e., Cutler et al. (2012) [22] proposed the UV-C dose of 1.21 mJ/cm$^2$ for 99.9% aerosolized PRRSV reduction (1 mJ = 0.001 J). In addition, the routine use of UV-C at 254 nm on farms raises concerns about its potentially harmful effects, such as eye and skin damages on both workers and animals.

More recently, reports on the inactivation of pathogens using "excimer" UV-C in vitro have appeared [24,25]. The excimer lamp (207–222 nm in the UV-C range) poses less risk to humans (innocuous to mammalian skins and eyes) compared with 254 nm [24], while still being germicidal for Methicillin-resistant *Staphylococcus aureus* (MRSA) [24] and H1N1 influenza virus [25]. There are no published data on the effectiveness of UV-C excimer (207–222 nm) to treat airborne PRRSV. Similarly, there are no published data on the inactivation of aerosolized PRRSV using UV-A (315–400 nm). UV-A is commonly known as "blacklight", considered less harmful, and used for, e.g., artificial sun tanning. Recent research has shown that both UV-C and UV-A can be effective in mitigating emissions of odor, odorous volatile organic compounds (VOCs), a potent greenhouse gas ($N_2O$), and ammonia ($NH_3$) [26–31] on lab and pilot scales. Thus, there is an opportunity to consider that UV treatment could be beneficial to mitigate both gaseous emissions and the pathogen load.

Given this background, we addressed specific objectives in this laboratory-scale proof-of-concept research:

1.  Quantify and compare the inactivation of aerosolized PRRSV by UV-C germicidal (254 nm), UV-C "excimer" (222 nm), UV-A fluorescent (365 nm), and UV-A LED (365 nm).
2.  Estimate the UV-A and UV-C dose needed for 90, 99, and 99.9% reduction in infectious aerosolized PRRSV.
3.  Evaluate the techno-economic feasibility of UV treatment for airborne PRRSV in a swine barn inlet air.

This research is a subsequent study following Li et al. (2021) [32] and Koziel et al. (2020) [33], where a system of PRRSV aerosolization, sampling, and recovery was designed, built, and verified.

## 2. Materials and Methods

### 2.1. Experiment Overview (Obj. 1)

The experimental setup (Figure 1) consisted of three major sections: (i) aerosolization section, (ii) treatment section, and (iii) sampling section, in the order of the airflow direc-

tion. Section (i) provided for PRRSV aerosolization, airflow regulation, and monitoring; Section (ii) provided the PRRSV aerosol-controlled UV exposure; Section (iii) sampled the aerosolized PRRSV after UV-treatment. The impinger liquid was tested for viable PRRSV, and the concentration was expressed as median tissue culture infectious dose (TCID$_{50}$).

The setup and procedure were based on Li et al. (2021) [32], except Section (ii), the treatment section, was installed with UV lamps to irradiate fast-moving PRRSV aerosols.

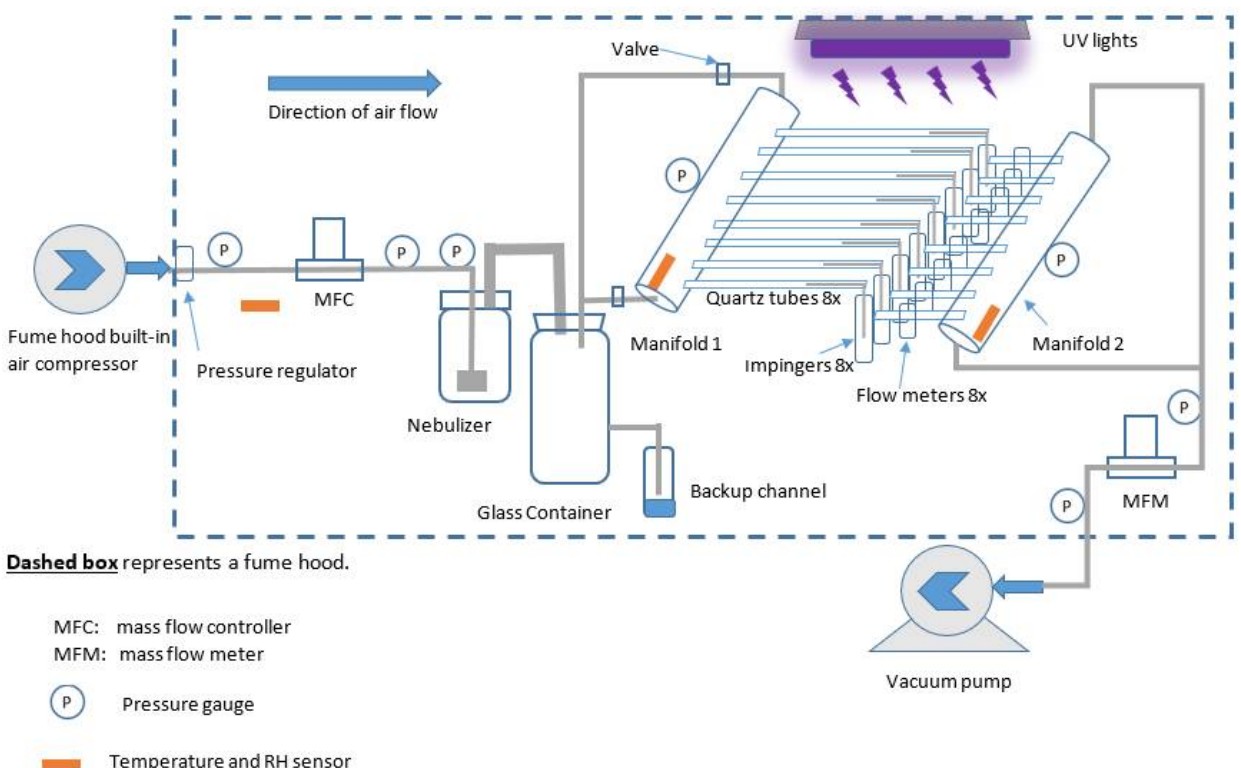

**Figure 1.** Experimental setup for porcine reproductive and respiratory syndrome virus (PRRSV) aerosol generation, UV treatment, and collection of treated PRRSV aerosols inside a fume hood.

Experimental procedures, e.g., PRRSV aerosolization and aerosol sampling, were conducted in a BSL-1 setting (Iowa State University, Ames, IA, USA), and the virus isolation was conducted in a BSL-2 laboratory (Veterinary Diagnostic Laboratory, College of Veterinary Medicine, Iowa State University).

*2.2. PRRSV Propagation and Aerosolization (Preparation for Obj. 1)*

The PRRSV used in this experiment (MN-184, PRRSV-2 Lineage 1) was provided by the Veterinary Diagnostic Laboratory (College of Veterinary Medicine, Iowa State University). The virus was propagated in the MARC-145 cell line that was derived from the African monkey kidney cell line MA-104 (Kim et al., 1993) [34]. The details of PRRSV propagation and storage are fully described in [32]. The propagated virus inoculum had a virus titer of $1 \times 10^{5.56}$ TCID$_{50}$/mL.

To generate PRRSV aerosols, compressed air first passed through a mass flow controller (MFC) (model#: GFCS-010013, Aalborg Instruments and Controls Inc., Orangeburg, New York, USA) and then to a 24-jet Collison nebulizer (formerly BGI Inc., now CH Technologies, Westwood, NJ, USA) [32]. The nebulizer was prefilled with 60 mL of PRRSV inoculum with 0.2% (*v/v*) antifoam A emulsion (Sigma-Aldrich Corp., St. Louis, MO, USA) and 2 ppm of Rhodamine B powder (Sigma-Aldrich Corp., St. Louis, MO, USA). The Rhodamine B served as a positive qualitative control to confirm the impingers' proper functioning to capture surviving aerosolized PRRSV. Antifoam A was previously shown to reduce foaming, with no harmful effect on laboratory cultured cells or PRRSV itself [35].

The nebulizer aerosolized ~35 mL of PRRSV inoculum under a pressure of around 20 psi in each (~45-min) experiment. PRRSV aerosols were directed to the glass container (3 gals, or 12 L) and then into Manifold 1, where they were distributed into eight branches (quartz tubes) for UV irradiation.

*2.3. UV Light Selections and Measurements (Obj. 1)*

2.3.1. UV Light Selections

Four types of UV light with three different wavelengths were used: (i) 365 nm UV-A fluorescent black light blue (BLB) light (F15T8BLB) (Ushio America Inc., Cypress, CA, USA), (ii) 365 nm UV-A LED lights (research-grade prototype, LED board + power supply) (Once Inc., Plymouth, MN, USA), (iii) 222 nm UV-C excimer light (Ushio America Inc., Cypress, CA, USA), and (iv) 254 nm UV-C germicidal light (G15T8) (Ushio America Inc., Cypress, CA, USA).

The UV-A fluorescent (365 nm) and UV-C germicidal (254 nm) bulbs were mounted in two XX-15A UV fixtures, respectively (Spectronics Corp., New York, NY, USA). Each fixture could accommodate two lamps, with each consuming 15 watts of power, although the full UV power is just a fraction of it. The LED lamp for UV-A (365 nm) was a square-shaped plate (0.3 × 0.45 m) mounted with dozens of small LED bulbs, equipped with an additional power supply. The UV-C excimer (222 nm) lamp was a single-bulb system with a customized power supply and its nominal power consumption of 300 W.

2.3.2. UV Light Intensity Measurement

The light intensity was measured (ILT 1700 radiometer, International Light Technologies, Peabody, MA, USA) using wavelength-specific sensors and filters. Four combinations of sensor/probe were used for the measurement of each lamp: (i) SED240 sensor added with an NS254 filter that can measure a center wavelength of 254 nm, with a tolerance range of 254–257 nm; (ii) SED240 sensor equipped with an NS220 filter that can measure a center wavelength of 220 nm, with a tolerance range of 219–223 nm; (iii) SED033 sensor with an NS365 filter that has a tolerance range of 365–367 nm (manufacturer information). The radiometer, all detectors, and filters were factory-calibrated before measurements. All UV lamps were turned on for 5 min before each measurement to ensure steady, stable, and consistent irradiation. During the light intensity measurements, the radiometer sensors were positioned 10 cm directly underneath the UV lights.

To better understand the UV intensity inside the quartz tube, the tubes were horizontally cut in half to allow the sensor to be positioned directly underneath the top half. The UV light intensity measurement was conducted on the layout of eight quartz tubes. Quartz tubes were used in this experiment as channels to carry and separate PRRSV aerosols. The circular, cross-sectional area of the tubes created difficulty when measuring and estimating UV light intensity and dose, as the UV sensor probe cannot be inserted inside for the real measurement. Therefore, we cut the quartz tube horizontally in half when measuring the light intensity. Seven equally spaced (~4 cm) points were measured over the length of each tube. In total, the light intensity at 56 points was measured, and an irradiation map was drawn for each type of UV. Figures A5–A8 in Appendix B show the maps of UV light intensity (irradiation) for all four types of UV lamps used in this experiment. Supplementary Figures S1–S7 present the UV lamps used in this experiment and an example of light intensity measurement.

During each experiment, the UV sensor was positioned at a reference point near Manifold 1 for monitoring purposes to ensure that the lamps were functioning normally. The summary of UV light intensity data was recorded in Table 1.

**Table 1.** Average UV light intensity for each treatment (quartz tube #) was measured without tube shielding. In the experiment, the quartz tubes were covered with different lengths of polyvinyl chloride (PVC) pipes to control the UV dose, and thus, the effective light intensity was estimated for each treatment separately.

| | UV Light Intensity (mW/cm$^2$) | | | |
|---|---|---|---|---|
| | **UV-C (254 nm)** | **UV-C (222 nm)** | **UV-A (365 nm, Fluorescent)** | **UV-A (365 nm, LED)** |
| Treatment 1 * | 3.43 | 1.10 | 0.57 | 1.71 |
| Treatment 2 | 4.40 | 1.36 | 0.70 | 1.93 |
| Treatment 3 | 5.04 | 1.49 | 0.77 | 2.00 |
| Treatment 4 | 5.35 | 1.49 | 0.80 | 2.02 |
| Treatment 5 | 5.35 | 1.41 | 0.79 | 2.01 |
| Treatment 6 | 5.06 | 1.33 | 0.74 | 1.99 |
| Treatment 7 | 4.53 | 1.19 | 0.66 | 1.90 |
| Treatment 8 | 3.57 | 1.04 | 0.49 | 1.66 |

* Treatment 1 refers to quartz tube 1. The same rule applies to all the treatments.

### 2.4. UV Treatment of Aerosolized PRRSV

Eight identical quartz tubes (ID = 25 mm, OD = 28 mm, length = 30 cm) (Technical Glass Products Inc., Painesville Twp., OH, USA) were positioned horizontally with an equal gap (5 cm) between each. Both ends of each tube were connected with plastic tubes and sealed by parafilm. The UV lights were placed on top of the tubes, 10 cm from the tubes' central horizontal plane. Additionally, note that the lamps were positioned horizontally at 90 degrees to the quartz tubes. This setting was intended to create a more uniform distribution of UV across the eight tubes.

Each treatment (quartz tube) represented a different level of UV dose. This was achieved by shielding different lengths of the quartz tubes with PVC tubes that were tailored to cover a desired section of the quartz, thereby achieving various treatment (UV exposure) times. The shielding options were 14.3, 28.6, 42.9, 57.2, 71.5, 85.8, 100% of the length of each quartz tube. The treatment times (calculated by dividing the exposed tube length by aerosol velocity) were 1.81, 1.55, 1.29, 1.03, 0.77, 0.52, 0.26, 0 s, i.e., from exposed to fully shielded. The layout of the treatments is shown in Figure A9 in Appendix B, and the illustrative photo is shown in Figure S7. For each type of UV, four (*n* = 4 replications) experiments were conducted.

The Bunsen–Roscoe Reciprocity Law dictates the calculation of the UV dose:

$$D = I \times T \tag{1}$$

where *D* is the UV dose (mJ/cm$^2$), *I* is irradiance or light intensity (mW/cm$^2$), and *T* is the UV treatment time (s).

### 2.5. Post-UV-Irradiation Aerosol Collection, Recovery, and PRRSV Titer Calculation (Obj. 1)

Before each experiment, the UV light was turned on for 5 min to stabilize UV irradiation. After that, the vacuum pump was turned on, immediately followed by opening the compressed air valve. The rear (right side) end of each quartz tube was connected to a glass AGI 7541 impinger (Ace Glass Inc., Vineland, NJ, USA) with a flow rate of 6 L/min. Each impinger was filled with 15 mL of PBS and 0.1% (*v/v*) antifoam A emulsion. The impingers function by impacting aerosolized virus onto the surface of the liquid. Thereafter, the liquid was tested to determine its concentration of infectious virus. Downstream of each impinger was a flow meter (Catalog No. RMA-21-SSV, Dwyer Instruments Inc., Michigan City, IN, USA) to adjust the flow rate to achieve equal distribution of the flow across the tubes. The sample collection time was 45 min per experiment. Desiccation of impinger

fluid is sometimes a problem in aerosol collection experiments. In this study, minimal impinger fluid loss was noted, i.e., <2 mL.

To determine the infectious virus titer, impinger fluid was transferred to a biosafety cabinet in a BSL-2 laboratory for 10-fold serial dilutions performed on 96-well plates, with eight replicates for each sample. Each well in 96-well plates (except for the first row) was prefilled with 270 μL of RPMI-1640 medium, and then, the sample was added to the plates' first row. The RPMI-1640 medium was supplemented with 10% fetal bovine serum, 2 mM L-glutamine, 0.05 mg/mL gentamicin, 100 unit/mL penicillin, 100 μg/mL streptomycin, and 0.25 μg/mL amphotericin [32]. The solution was then mixed, and 30 μL of liquid was transferred sequentially from one row to another. Thus, the dilution for each row ranged from $10^0$, $10^{-1}$, . . . , to $10^{-7}$, respectively. Thereafter, 100 μL from each well was inoculated into subconfluent MARC-145 cells grown in 96-well plates. The plates were incubated at 37 °C in a humidified 5% $CO_2$ incubator. Cytopathic effect (CPE) development was checked under an optic microscope daily, and infected wells were marked as positive until no additional wells were identified as infected (5 to 7 days). To confirm the presence of PRRSV, the cell plates were fixed (80% acetone for 10 min), dried, and then stained with a PRRSV nucleocapsid protein-specific monoclonal antibody (SDOW17-F) conjugated to fluorescent isothiocyanate (Rural Technologies, Inc., Brookings, SD, USA) for 1 h at 37 °C in the incubator. The antibody conjugate was decanted, and the cell plates were washed with PBS (1×, pH 7.4) 3 times, 5 min each time. Plates were read under an Olympus IX71 fluorescent microscope (Olympus America Inc., Center Valley, PA, USA) [32]. The Spearman–Kärber method [36–38] was used to calculate the virus titers based on the number of wells showing positive PRRSV-specific fluorescence at specific dilution, and the results were expressed as $TCID_{50}$/mL of the impinger sampling fluid.

*2.6. UV Inactivation Models (Obj. 2)*

The statistical analysis was conducted using the *R* statistical program (version 3.6.3, R Studio, Boston, MA, USA). The packages used for modeling were "car" (for model comparison and lack of fit) [39], and "minpack.lm" [40] and "nlme" [41] (both for two-stage models). The inactivation models shown in the Results section were based on the one-stage and two-stage microbial survival models mentioned in Kowalski (2000) [42] and Riley et al. (1972) [43]. Briefly, the one-stage microbial inactivation (or reduction) model treated the microbial population as a homogeneous population in which all members are equally susceptible to inactivation, i.e., a monomolecular reaction [42]. In contrast, two-stage inactivation assumes the microbial population is composed of two types, one type more susceptible to UV and one less so. Cutler et al. (2011) [23] analyzed the fitness of one-stage and two-stage inactivation models for static PRRSV inoculum. In this research, the data were processed in both one- and two-stage inactivation models.

The one-stage model (adopted from Chick's law),

$$log_{10}N_t = log_{10}N_0 - kD_t + C_1 \tag{2}$$

Rearrange Equation (2) to get the fraction of surviving pathogens ($log_{10}$ normalized),

$$log_{10}\frac{N_t}{N_0} = -kD_t + C_1 \tag{3}$$

where

$N_t$ = virus titer ($TCID_{50}$/mL) in the impinger fluid after UV treatment with a $D_t$;
$N_0$ = virus titer ($TCID_{50}$/mL) in the control sample (without UV exposure);
$k$ = inactivation rate ($cm^2$/mJ);
$D_t$ = UV dose (mJ/$cm^2$), calculated by Equation (1);
$C_1$ = intercept for the one-stage model.

The two-stage model can be expressed as,

$$log_{10}N_t = log_{10}N_0 + log_{10}\left[(1-f)\cdot 10^{-k_1 \cdot D_t} + f \cdot 10^{-k_2 \cdot D_t}\right] + C_2 \tag{4}$$

Rearrange Equation (4) to get the fraction of surviving pathogens ($log_{10}$ normalized),

$$log_{10}\frac{N_t}{N_0} = log_{10}\left[(1-f)\cdot 10^{-k_1 \cdot D_t} + f \cdot 10^{-k_2 \cdot D_t}\right] + C_2 \tag{5}$$

$1 - f$ = the fraction of the virus population that is more susceptible to UV treatment with an inactivation rate $k_1$;
$f$ = the fraction of the virus population that is more resistant to UV treatment with an inactivation rate $k_2$;
$k_1$ = inactivation rate for the susceptible fraction of the virus population under UV treatment;
$k_2$ = inactivation rate for the resistant fraction of the virus population under UV treatment;
$C_2$ = intercept for the two-stage model.

## 3. Results

### 3.1. Effectiveness of UV to Treat Airborne PRRSV (Obj. 1)

Figures 2–5 summarize the UV inactivation of aerosolized PRRSV. Two curves (one-stage and two-stage) were drawn to show the data fit in the inactivation models, which described the kinetics of inactivation. We adopted the one-stage and two-stage microbial survival models previously described by Kowalski et al. (2000) [42] and Riley et al. (1972) [43]. The following is an analysis of our results for the purpose of modeling the UV doses required for PRRSV inactivation.

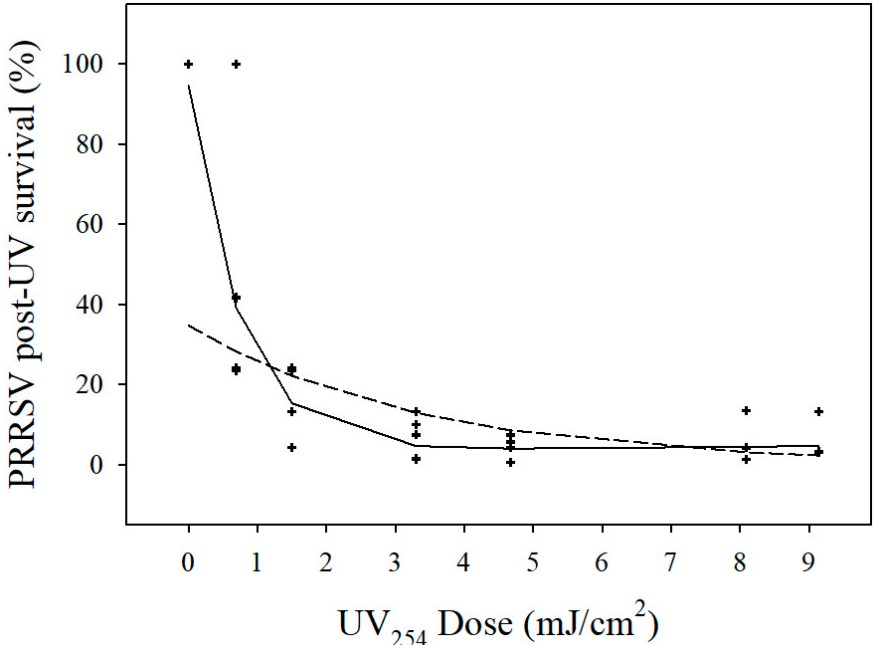

**Figure 2.** UV-C (254 nm) treatment inactivation of aerosolized PRRSV. PRRSV post-UV survival (%) = $N_t/N_0$. A log10 normalized PRRSV post-UV survival is shown in Figure A1. One-stage and two-stage inactivation models are marked with dashed and solid lines, respectively.

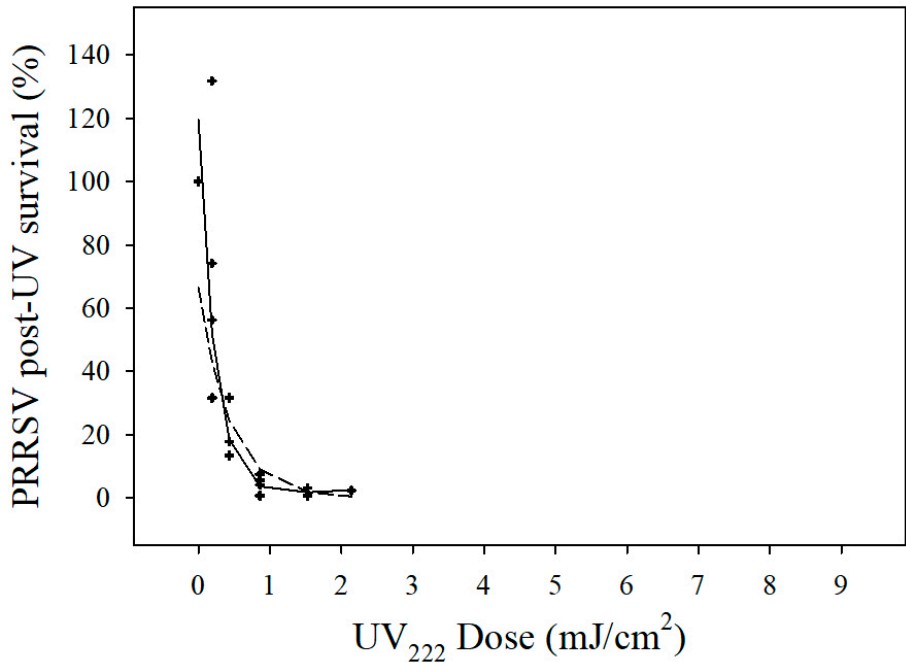

**Figure 3.** UV-C (222 nm) treatment inactivation of aerosolized PRRSV. PRRSV post-UV survival (%) = $N_t/N_0$. A $\log_{10}$ normalized PRRSV post-UV survival is shown in Figure A2. One-stage and two-stage inactivation models are marked with dashed and solid lines, respectively.

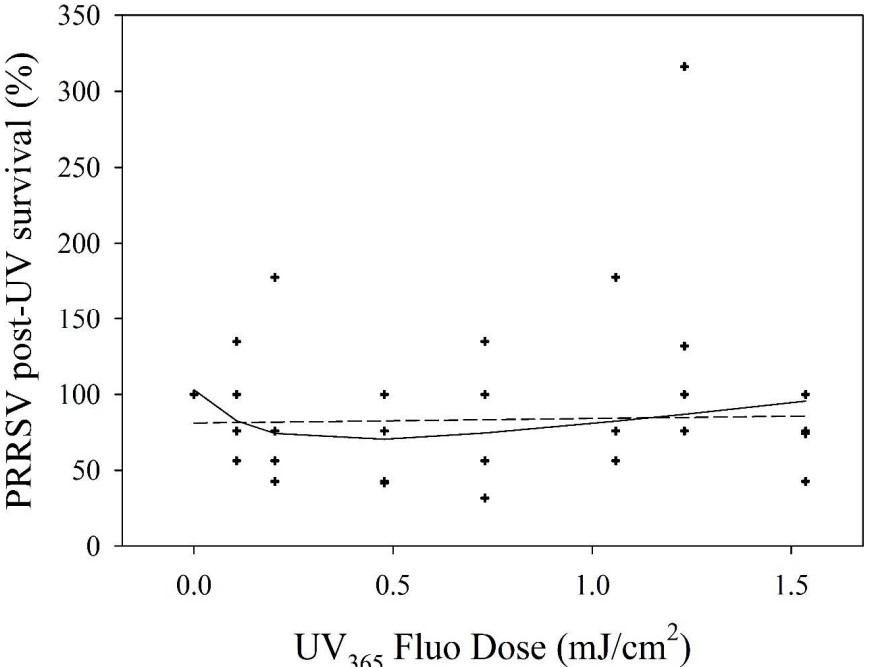

**Figure 4.** UV-A (365 nm, fluorescent) treatment inactivation of aerosolized PRRSV. PRRSV post-UV survival (%) = $N_t/N_0$. A $\log_{10}$ normalized PRRSV post-UV survival is shown in Figure A3. One-stage and two-stage inactivation models are marked with dashed and solid lines, respectively.

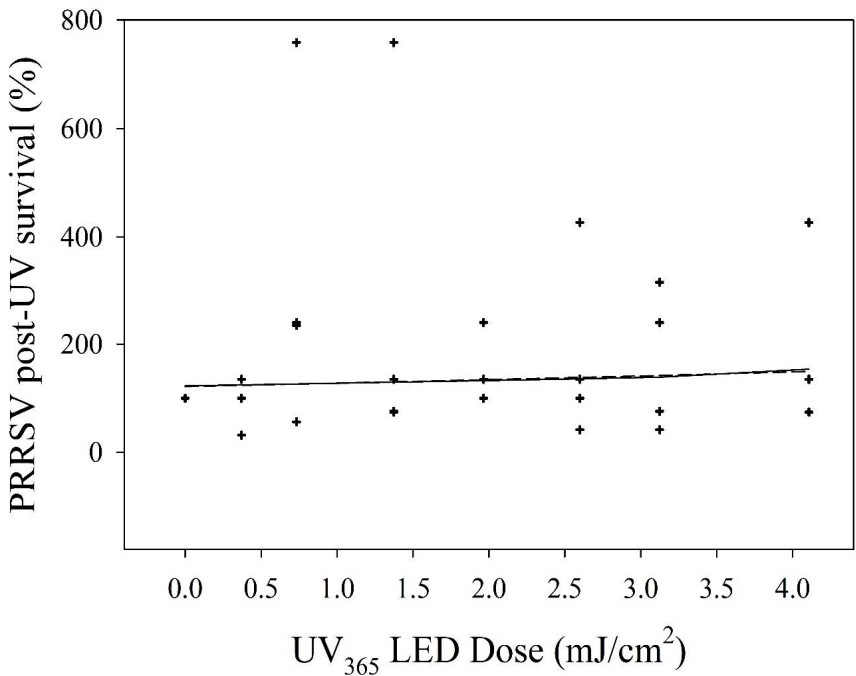

**Figure 5.** UV-A (365 nm, LED) treatment inactivation of aerosolized PRRSV. PRRSV post-UV survival (%) = $N_t/N_0$. A $\log_{1supp0}$ normalized PRRSV post-UV survival is shown in Figure A4. One-stage and two-stage inactivation models are marked with dashed and solid lines, respectively.

### 3.2. Estimations of UV Dose Needed for 90, 99, and 99.9% Airborne PRRSV Reduction (Obj. 2)

Table 2 shows the parameters (including inactivation rate, *k*) of the one-stage and two-stage models derived from the experimental data. The raw experimental data was organized into a spreadsheet (UV dose vs. PRRSV titer.xlsx) included in Supplementary Materials.

**Table 2.** One-stage and two-stage models, parameter estimations, and model parameters for inactivation of airborne PRRSV with four types of UV light tested.

| Parameters / UV Types | UV-C (254 nm) | UV-C Excimer (222 nm) | UV-A (365 nm, Fluorescent) | UV-A (365 nm, LED) |
|---|---|---|---|---|
| **Two-stage inactivation model** | | | | |
| Intercept | −0.02367 | 0.0775 | 0.012422 | −0.02748 |
| Susceptible virus population fraction (*f*) | 0.9675 | 0.9951 | 0.57417 | 1.72542 |
| Resistant virus population fraction (1 − *f*) | 0.0325 | 0.0049 | 0.42583 | −0.72542 |
| Inactivation rate (constant), $k_1$, for resistant virus population ($cm^2/mJ$) | 0.01927 | 0.3014 | −0.13585 | 0.88296 |
| Inactivation rate (constant), $k_2$, for susceptible virus population ($cm^2/mJ$) | −0.58947 | −1.9189 | 2.89537 | 0.02075 |
| Lack-of-fit test *p*-value | *p* = 0.8488 | *p* = 0.6565 | *p* = 0.2848 | *p* = 0.6532 |
| **One-stage inactivation model** | | | | |
| Intercept | −0.4601 | −0.1779 | −0.09103 | 0.08388 |
| Inactivation rate, *k* ($cm^2/mJ$) | −0.1307 | −1.0053 | 0.01556 | −0.02183 |
| Lack-of-fit test *p*-value | *p* = 0.0153 | *p* = 0.01344 | *p* = 0.3336 | *p* = 0.7748 |

Both inactivation curves of UV-C germicidal (254 nm) and UV-C excimer (222 nm) fit better (with lack-of-fit *p*-value > 0.1) with a two-stage inactivation model rather than a one-stage inactivation model, with lack-of-fit *p*-value = 0.8488 and 0.6565, respectively. This finding is consistent with previous research on PRRSV [23]. It was not within this study's scope to determine why two-stage fits the data better than one-stage—that would be a different study and challenging to execute.

After incorporating the parameters from Table 2 to Equations (3) and (5), the model equations for each type of UV are expressed as follows.

For UV-C (254 nm) (data shown in Figure 2), one-stage and two-stage, respectively,

$$log_{10}\frac{N_t}{N_0} = -0.1307 \cdot D_t - 0.4601 \tag{6}$$

$$log_{10}\frac{N_t}{N_0} = log_{10}\left[0.9675 \cdot 10^{-0.58947 \cdot D_t} + 0.0325 \cdot 10^{0.01927 \cdot D_t}\right] - 0.02367 \tag{7}$$

For UV-C (222 nm) (data shown in Figure 3), one-stage and two-stage, respectively,

$$log_{10}\frac{N_t}{N_0} = -1.0053 \cdot D_t - 0.1779 \tag{8}$$

$$log_{10}\frac{N_t}{N_0} = log_{10}\left[0.9951 \cdot 10^{-1.9189 \cdot D_t} + 0.0049 \cdot 10^{0.3014 \cdot D_t}\right] + 0.0775 \tag{9}$$

For UV-A (365 nm, fluorescent) (data shown in Figure 4), one-stage and two-stage, respectively,

$$log_{10}\frac{N_t}{N_0} = -0.01556 \cdot D_t - 0.09103 \tag{10}$$

$$log_{10}\frac{N_t}{N_0} = log_{10}\left[0.42583 \cdot 10^{0.13585 \cdot D_t} + 0.57417 \cdot 10^{-2.89537 \cdot D_t}\right] + 0.012422 \tag{11}$$

For UV-C (365 nm, LED) (data shown in Figure 5), one-stage and two-stage, respectively,

$$log_{10}\frac{N_t}{N_0} = 0.02183 D_t + 0.08388 \tag{12}$$

$$log_{10}\frac{N_t}{N_0} = log_{10}\left[-0.72542 \cdot 10^{-0.88296 \cdot D_t} + 1.72542 \cdot 10^{-0.02075 \cdot D_t}\right] - 0.02748 \tag{13}$$

Table 3 summarized the estimated or projected UV doses for target percentage (%) PRRSV reduction of 90, 99, and 99.9%, based on one-stage and two-stage models.

**Table 3.** Estimated the UV dose (mJ/cm$^2$) needed for target % aerosolized PRRSV reduction in fast-moving air, using both one-stage and two-stage inactivation models.

|  | 90% (1-log) Reduction | | 99% (2-log) Reduction | | 99.9% (3-log) Reduction | |
|---|---|---|---|---|---|---|
|  | **1-Stage** | **2-Stage** | **1-Stage** | **2-Stage** | **1-Stage** | **2-Stage** |
| UV-C (254 nm) | 4.131 | 1.933 | 11.782 | 2.662 | 19.433 [a] | 2.442 |
| UV-C (222 nm) | 0.818 | 0.581 | 1.812 | 1.036 | 2.807 | 1.044 |
| UV-A (365 nm, fluor.) | 58.42 | - | 122.684 | - | 186.952 | - |
| UV-A (365 nm, LED) | −49.651 [b] | 58.285 | −95.460 [b] | 106.478 | −141.268 [b] | 154.671 |

[a] UV-C (254 nm) dose needed to inactivate 99.9% aerosolized PRRSV was estimated to be 1.21 mJ/cm$^2$ by Cutler et al., 2012 [22]. [b] Negative values are not considered biologically meaningful, i.e., the UV light under these categories did not have an inactivation effect for the doses used in the experiment.

### 3.3. Preliminary Techno-Economic Analysis of Potential Farm-Scale Application (Obj. 3)

Table 4 shows the comparison of electricity cost (operational cost) among four different UV light types used for this laboratory-scale experiment.

**Table 4.** Comparison of electricity cost (operational cost) among four different UV light types used for this laboratory-scale experiment.

| UV Light | Measured Power Consumption (W) [a] | Electricity Consumption (kWh) [b] | Electricity Cost [c] | Cost of UV Lamps [d] |
|---|---|---|---|---|
| UV-C (254 nm) | 50.5 | 0.038 | USD 0.0023 | <USD 100 |
| UV-C (222 nm) | 250 | 0.19 | USD 0.0122 | ~USD 600 |
| UV-A (365 nm, fluor.) | 49.5 | 0.037 | USD 0.0022 | <USD 100 |
| UV-A (365 nm, LED) | 43.8 | 0.033 | USD 0.0019 | ~USD 200 [e] |

[a] The total power consumption of all lamps used for each experiment, i.e., 4 bulbs for UV-C (254 nm) and UV-A (fluorescent), respectively; 1 lamp for UV-C excimer, and 1 lamp for UV-A (LED); [b] the percentage of effective UV irradiated area (on PRRSV) with respect to its total irradiation area (estimated); [c] electricity cost in rural areas (USA Midwest) = USD 0.12/kWh; [d] cost of UV lamps and fixtures, excluding other experimental devices, as described in the Material and Methods section; [e] cost estimation of a research-grade prototype of the LED lamp (LED board + power supply).

Table 5 is a summary of preliminary cost estimation of UV application vs. HEPA filtration system.

**Table 5.** Estimations of the cost of implementing UV-C (254 nm) light or HEPA filtration treatment in a 1000-head swine barn with different swine types for 1 year. Estimations were based on extrapolations from this laboratory-scale study.

| Type | Capital Cost (Hot Weather [a]) | 1-Year Electricity Cost (Mixed Weather [a]) | Maintenance | Total Cost |
|---|---|---|---|---|
| UV light with pre-filters | USD 66,000 | USD 35,000 | USD 6600 | USD 107,600 |
| HEPA filters with pre-filters | USD 80,400 | N/A | USD 8040 | USD 88,440 |

[a] Estimations and assumptions on the size of the farm, ventilation rates, and weather conditions are from MWPS-8 "Swine Housing and Equipment Handbook" [44].

## 4. Discussion

### 4.1. UV Effectiveness and Inactivation Models

To date, most of the research on ultraviolet inactivation of pathogens has been under static conditions, i.e., no viral aerosol generation or flow under UV irradiation [21]. Only a fraction of such research involved dynamic targets, i.e., fast-moving aerosols rather than stationary cell plates or Petri dishes. There are much more UV irradiation experiments on stationary (static) objects than on dynamic (flowing) targets, so is the case for PRRSV treatment. To date, Cutler et al. (2012) [22] was the benchmark on UV irradiation on aerosolized PRRSV. Other research experiments focused on UV inactivation of PRRSV on stationary objects, e.g., on tissue culture plates [23], on common farm surfaces (rubber, concrete, paper, etc.) [15], on samples inside irradiation chamber [45].

In most cases, the UV doses cumulated in static systems were much higher than in dynamic systems because aerosols' flow significantly reduces the irradiation (contact) time on the targets (bacteria, viruses, etc.). However, according to Kowalski et al. (2000) [42], the UV inactivation rates tend to be higher in dynamic conditions than in static conditions, and thus, less dose is needed. It was speculated that pathogens flowing and tumbling in the air can receive UV irradiation all around, while under static conditions, the exposure is only directed in one plane or side, and thus, it is less efficient.

The theoretical minimum titer determined using the Spearman–Kärber method was $1 \times 10^{0.5}$ (or 0.5 $\log_{10}$) $TCID_{50}$/mL. In this experiment, the detected PRRSV titer values were all above $1 \times 10^{1.5}$ $TCID_{50}$/mL, and the control samples (from Treatment 5) had a virus titer of about ~$1 \times 10^4$ $TCID_{50}$/mL. This research showed that this UV chamber achieved ~2-log reduction in aerosolized PRRSV with a UV dose <5 mJ/cm$^2$ (UV-C, 254 nm), or <2 mJ/cm$^2$ (UV-C, 222 nm), under experimental conditions. The experiment itself did not achieve a higher level of PRRSV titer reduction. Considering this, the estimation of 1-log and 2-log airborne PRRSV reductions was reasonable and within the data scope, while estimation for 3-log reduction would be less accurate than the former two. Thus, the model was extrapolated to estimate the dose needed for a higher reduction level (i.e., >2-log). The

model's accuracy could be improved if the data represented a wider range of UV doses inactivating higher PRRSV titers. To estimate a UV dose required for a higher log reduction would require a higher initial concentration of virus load to start with. Extending the sampling time may increase the initial PRRSV titer, but a very long sampling time may reduce virus viability before titration and, thus, reduce the virus titer's accuracy derived from cell culture plates.

The experimental data were evaluated using both one-stage and two-stage fit with inactivation models for all four types of UV light used in this experiment. The two-stage model provided a better fit with both UV-C 254 nm and UV-C 222 nm, with lack-of-fit *p*-values of 0.8488 and 0.6565 (both >0.1), respectively. Due to the magnitude difference of doses needed for 3-log reduction between one-stage and two-stage models, 19.43 and 2.44 mJ/cm$^2$, respectively, we reported both values for consideration, but 2.44 mJ/cm$^2$ is more realistic and similar to the dose 1.21 mJ/cm$^2$ reported by Cutler et al. (2012) [22]. The estimated UV-C (222 nm) doses were 2.81 and 1.04 mJ/cm$^2$ for one-stage and two-stage models, respectively.

For UV-A (365 nm) (both fluorescent and LED-based), the reduction in PRRSV titer was not found for the dose up to 4.11 mJ/cm$^2$. This experiment does not rule out UV-A's effectiveness, but a much higher UV dose may be needed to achieve the same log reduction or to have a significant germicidal effect.

### 4.2. Exploring the UV Inactivation Mechanism

UV light inactivates pathogens by causing the DNA or RNA structure to distort, and thus, a normal replication cannot happen. Specifically, UV light may cause the formation of pyrimidine dimers between two adjacent or opposing pyrimidines. The dimers may eventually result in breaks in the genome by affecting the sugar backbone [46,47]. The inactivation of the RNA by the UV-induced uracil dimers could also affect the RNA to serve as a transcription template [48]. It is noteworthy that UV-C has a stronger ability to induce dimer formation than UV-B, followed by UV-A [49]. UV does not directly eliminate the pathogen (bacteria or viruses); at least, this is not its primary means of inactivation, although, at a higher dose, this may happen by speculation. Thus, to understand and verify the mechanism of UV irradiation on PRRSV, an additional PCR test could be added in this experiment. If the PCR results show no significant reduction while bioassay does, that could help confirm the mechanism of UV inactivation to some extent.

### 4.3. Techno-Economic Analysis

Per feasibility evaluation, due to the high cost of UV-C excimer (in the magnitude of thousand USD per high-power lamp), the economic analysis was only conducted for generic UV-C (254 nm) lamps to consider its feasibility for farm-scale application. To date, the cost of UV-C excimer lamps is higher than the cost of generic UV-C (254 nm). Cost comparison for a 1000-head swine barn, 1-year maintenance was conducted between the UV light and HEPA filtration systems. The estimation indicated that UV-C light cost is estimated to be USD 107,600, while HEPA filtration costs approximately USD 88,400. The cost difference is within 25%. This filtration cost estimate did not include the capital cost, estimated at ~USD 150–200 per sow (or ~USD 450,000–600,000 for 3000-sow herd) as reported by Alonso et al., 2013 [50]. However, UV lamps usually need to be replaced after 8000 h (<1-y) of operation, while HEPA filters typically last for a few years if maintained well. On the other hand, the UV light system is less labor-intensive for monitoring and replacements. The UV light can be turned off when the risk is low, thus allowing farmers to make cost-conscious decisions. Based on these two factors, the cost comparison for a long-term implementation deserves more data collection and study to lead to a more data-driven metric.

## 5. Conclusions

The results show that UV-C (254 nm) and UV-C excimer (222 nm) effectively inactivated aerosolized PRRSV ~99% (2-log) with a dose <5 mJ/cm$^2$ for UV-C (254 nm) dose, or <2 mJ/cm$^2$ for UV-C (222 nm). UV doses needed for 1-log, 2-log, and 3-log virus titer reductions were estimated using both one-stage and two-stage inactivation models. For both types of UV-A (365 nm) lamps, no reduction in PRRSV titer was found with the UV dose used in this experiment. This experiment does not rule out UV-A effectiveness, but a much higher UV dose may be needed. Preliminary economic analysis showed that UV light costs the same magnitude as HEPA filtration systems in terms of materials and electricity at a farm-scale implementation. However, more data and research are needed to make accurate and long-term predictions.

**Supplementary Materials:** The following are available online at https://www.mdpi.com/2077-04 72/11/3/259/s1. Supplementary Figures S1–S6 show the UV lamps used in this experiment, as well as demonstrations of measuring light intensity. Figure S1. UV-C (254 nm) germicidal lamps used in this experiment. Figure S2. UV-A (365 nm) fluorescent BLB (blacklight blue) lamps were used in this experiment. Figure S3. UV-C (222 nm) excimer lamp were used in this experiment. Figure S4. UV-A (365 nm) LED lamp was used in this experiment. Figure S5. This figure shows the setup of UV light intensity measurement. A UV sensor probe was covered by a quartz tube's semi-circular shape to simulate the light intensity in the middle plane of the quartz tube. Figure S6. This figure shows the setup of UV light intensity (irradiance) measurement. The operator wears a UV-proof face shield while measuring the UV light intensity. Another operator is recording the data from the radiometer screen (not shown in this photo). Figure S7. This photo shows the setup to control the UV dose by short sections of PVC shielding the quartz tubes with the PRRSV aerosol from irradiation. A well-organized Excel spreadsheet (UV dose vs. PRRSV titer corrected.xlsx) is provided, including UV doses and infectious PRRSV titer data.

**Author Contributions:** Conceptualization, J.A.K., J.J.Z., S.J.H. and W.S.J.; methodology, J.A.K., J.J.Z., J.Z., P.L., T.-Y.C. and W.Y.-I.; validation, P.L., J.A.K., J.J.Z., T.-Y.C., M.L. and B.C.; formal analysis, T.-Y.C. and P.L.; investigation P.L. and T.-Y.C.; resources P.L., J.A.K., T.-Y.C., W.Y.-I., J.J.Z. and J.Z.; data curation P.L. and T.-Y.C.; writing—original draft preparation, P.L.; writing—review and editing, J.A.K., J.J.Z., J.Z., W.S.J. and P.L.; visualization, P.L.; supervision, J.A.K., J.J.Z. and J.Z.; project administration, J.A.K. and J.J.Z.; funding acquisition, J.A.K., J.J.Z., J.Z., W.S.J. and S.J.H. All authors have read and agreed to the published version of the manuscript.

**Funding:** This project was funded by checkoff dollars through the National Pork Board (NPB). (Project Title: Mitigation of PRRS transmission with UV light treatment of barn inlet air: proof-of-concept. Project number: NPB #18-160. In addition, this research was partially supported by the Iowa Agriculture and Home Economics Experiment Station, Ames, Iowa. Project no. IOW05556 (Future Challenges in Animal Production Systems: Seeking Solutions through Focused Facilitation) sponsored by the Hatch Act and State of Iowa funds (J.A.K.).

**Institutional Review Board Statement:** The Institutional Biosafety Committee (IBC) of Iowa State University approved the study on 14 April 2020. Reference number: IBC-20-016. Title: Mitigation of PRRS transmission with UV light treatment of barn inlet air: proof-of-concept. An amendment to the protocol was approved on 24 August 2020.

**Data Availability Statement:** The raw data are available in a spreadsheet (UV dose vs. PRRSV titer corrected.xlsx), which is accessible at Supplementary Materials.

**Acknowledgments:** The authors are very thankful to Holger Claus and Ryan Olsen (Ushio America Inc.) for pointing out the discrepancy in reported UV-C 254 nm and 222 nm lamp outputs that led us to investigate, find the source of discrepancy, and correct the manuscript.

**Conflicts of Interest:** The authors declare no conflict of interest. The funders had no role in the design of the study, in the collection, analyses, or interpretation of data, in the writing of the manuscript, or in the decision to publish the results.

## Appendix A

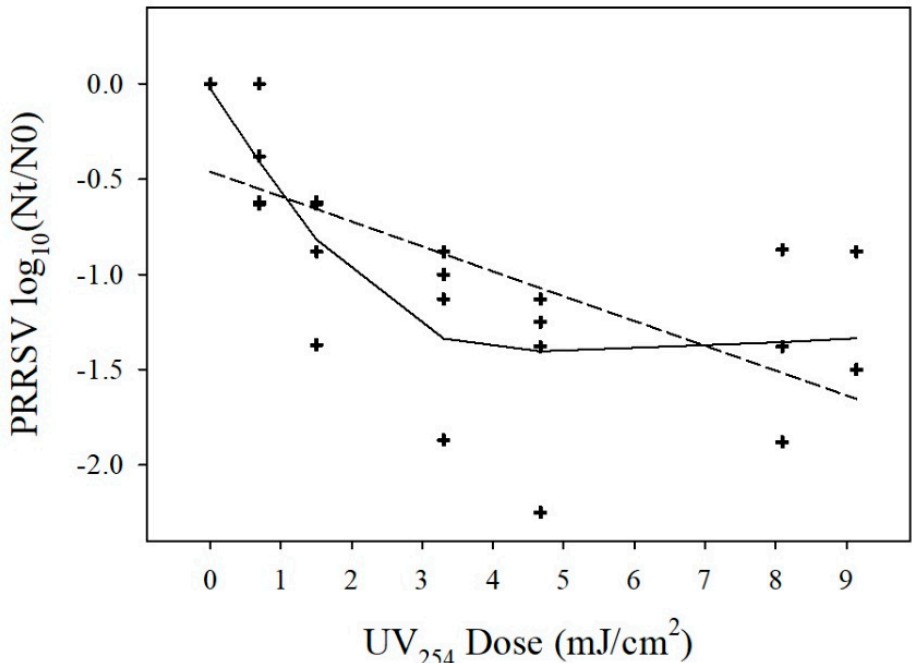

**Figure A1.** UV-A (254 nm) treatment inactivation curve on aerosolized PRRSV. One-stage (dashed line) and two-stage (solid line) inactivation curves were drawn. PRRSV survival is expressed as $\log_{10}(N_t/N_0)$, not as a percentage (%) as shown in Figure 2.

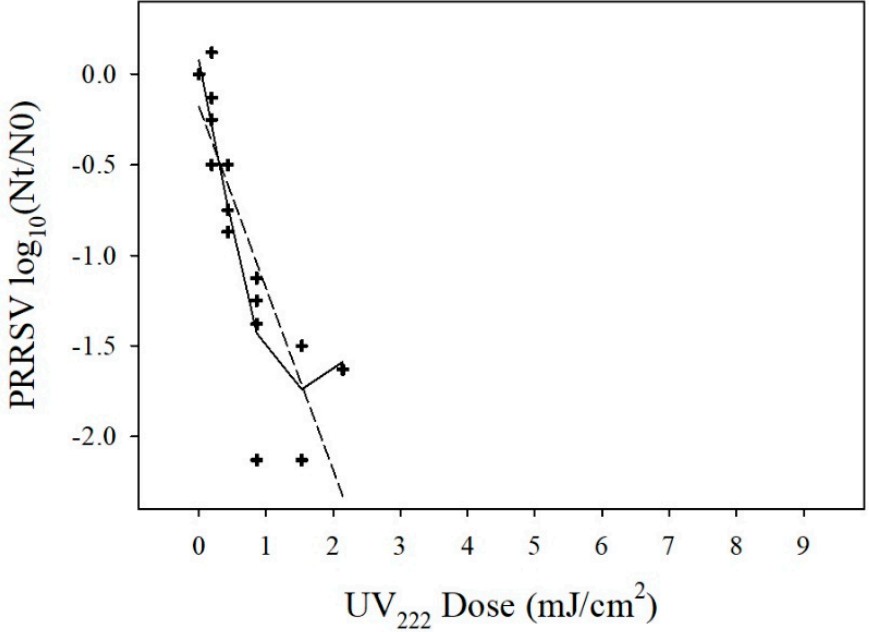

**Figure A2.** UV-C (222 nm) treatment inactivation curve on aerosolized PRRSV. One-stage (dashed line) and two-stage (solid line) inactivation curves were drawn. PRRSV survival is expressed as $\log_{10}(N_t/N_0)$, not as a percentage (%) as shown in Figure 3.

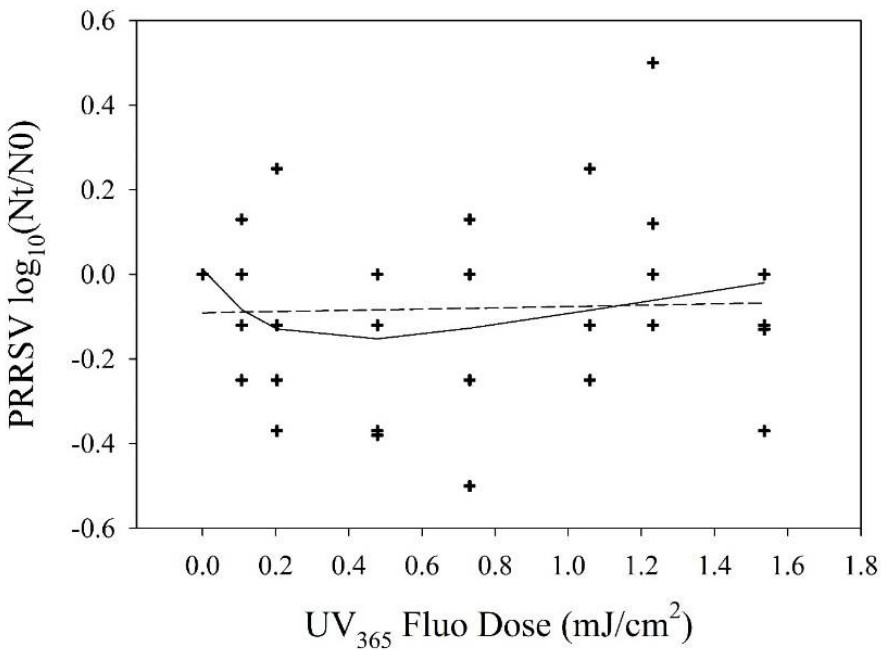

**Figure A3.** UV-A (365 nm, fluorescent) treatment inactivation curve on aerosolized PRRSV. One-stage (dashed line) and two-stage (solid line) inactivation curves were drawn. PRRSV survival is expressed as $\log_{10}(N_t/N_0)$, not as a percentage (%) as shown in Figure 4.

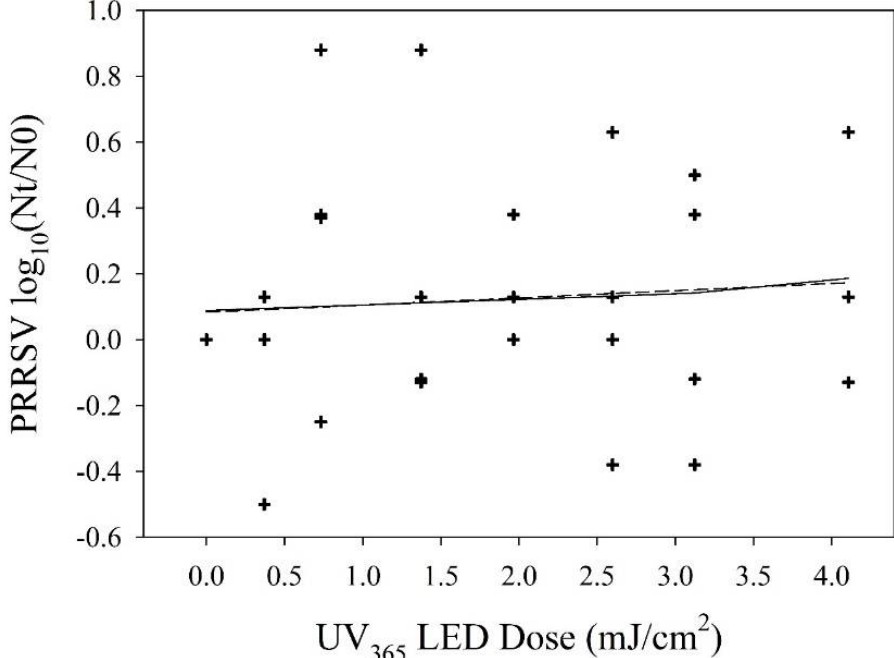

**Figure A4.** UV-A (365 nm, LED) treatment inactivation curve on aerosolized PRRSV. One-stage (dashed line) and two-stage (solid line) inactivation curves were drawn. PRRSV survival is expressed as $\log_{10}(N_t/N_0)$, not as a percentage (%) shown in Figure 5.

## Appendix B

| Treatment 1 | 2.66 | 3.27 | 4.05 | 4.21 | 4.11 | 3.12 | 2.59 |
|---|---|---|---|---|---|---|---|
| Treatment 2 | 3.41 | 4.13 | 5.39 | 5.19 | 5.20 | 4.09 | 3.37 |
| Treatment 3 | 3.84 | 4.75 | 6.01 | 5.98 | 6.03 | 4.76 | 3.92 |
| Treatment 4 | 3.99 | 5.00 | 6.37 | 6.29 | 6.53 | 5.11 | 4.13 |
| Treatment 5 | 3.89 | 4.91 | 6.37 | 6.36 | 6.55 | 5.14 | 4.25 |
| Treatment 6 | 3.66 | 4.66 | 6.02 | 5.99 | 6.13 | 4.86 | 4.09 |
| Treatment 7 | 3.25 | 4.15 | 5.39 | 5.36 | 5.49 | 4.40 | 3.65 |
| Treatment 8 | 2.45 | 3.36 | 4.41 | 4.28 | 4.26 | 3.33 | 2.92 |

**Figure A5.** Map of UV light intensity (irradiance) measurements for UV-C (254 nm) for Treatment 1 to Treatment 8. Each colored grid represents a ~4 × 4 cm area where the UV sensor was in place for measurement. The color gradient green–yellow–red represents increasing UV light intensity. Units: mW/cm$^2$.

| Treatment 1 | 0.74 | 1.04 | 1.41 | 1.55 | 1.34 | 0.98 | 0.67 |
|---|---|---|---|---|---|---|---|
| Treatment 2 | 0.87 | 1.28 | 1.78 | 1.99 | 1.64 | 1.17 | 0.82 |
| Treatment 3 | 0.91 | 1.38 | 1.92 | 2.17 | 1.91 | 1.25 | 0.90 |
| Treatment 4 | 0.96 | 1.37 | 1.92 | 2.14 | 1.89 | 1.26 | 0.92 |
| Treatment 5 | 0.90 | 1.32 | 1.78 | 1.98 | 1.78 | 1.26 | 0.87 |
| Treatment 6 | 0.84 | 1.22 | 1.70 | 1.90 | 1.68 | 1.17 | 0.79 |
| Treatment 7 | 0.72 | 1.10 | 1.50 | 1.67 | 1.52 | 1.08 | 0.74 |
| Treatment 8 | 0.63 | 1.02 | 1.36 | 1.43 | 1.27 | 0.93 | 0.61 |

**Figure A6.** Map of UV light intensity (irradiance) measurements for UV-C (222 nm) for Treatment 1 to Treatment 8. The color gradient green–yellow–red represents increasing UV light intensity. Units: mW/cm$^2$.

| | | | | | | | |
|---|---|---|---|---|---|---|---|
| Treatment 1 | 0.42 | 0.55 | 0.64 | 0.67 | 0.66 | 0.61 | 0.45 |
| Treatment 2 | 0.49 | 0.73 | 0.80 | 0.80 | 0.78 | 0.75 | 0.57 |
| Treatment 3 | 0.54 | 0.81 | 0.86 | 0.87 | 0.88 | 0.82 | 0.63 |
| Treatment 4 | 0.54 | 0.84 | 0.90 | 0.91 | 0.92 | 0.84 | 0.64 |
| Treatment 5 | 0.50 | 0.83 | 0.89 | 0.90 | 0.91 | 0.85 | 0.65 |
| Treatment 6 | 0.47 | 0.77 | 0.84 | 0.84 | 0.85 | 0.81 | 0.63 |
| Treatment 7 | 0.42 | 0.69 | 0.75 | 0.74 | 0.76 | 0.71 | 0.56 |
| Treatment 8 | 0.29 | 0.50 | 0.60 | 0.59 | 0.55 | 0.49 | 0.39 |

**Figure A7.** Map of UV light intensity (irradiance) measurements for UV-A (365 nm, fluorescent) for Treatment 1 to Treatment 8. The color gradient green–yellow–red represents increasing UV light intensity. Units: mW/cm$^2$.

| | | | | | | | |
|---|---|---|---|---|---|---|---|
| Treatment 1 | 1.44 | 1.60 | 1.89 | 1.96 | 1.91 | 1.70 | 1.50 |
| Treatment 2 | 1.51 | 1.91 | 2.08 | 2.12 | 2.09 | 2.01 | 1.81 |
| Treatment 3 | 1.51 | 1.96 | 2.16 | 2.21 | 2.20 | 2.07 | 1.87 |
| Treatment 4 | 1.47 | 1.93 | 2.18 | 2.24 | 2.25 | 2.12 | 1.91 |
| Treatment 5 | 1.45 | 1.92 | 2.16 | 2.23 | 2.24 | 2.15 | 1.90 |
| Treatment 6 | 1.50 | 1.92 | 2.13 | 2.20 | 2.22 | 2.13 | 1.83 |
| Treatment 7 | 1.42 | 1.86 | 2.04 | 2.09 | 2.12 | 2.03 | 1.72 |
| Treatment 8 | 1.19 | 1.65 | 1.87 | 1.88 | 1.79 | 1.70 | 1.52 |

**Figure A8.** Map of UV light intensity (irradiance) measurements for UV-C (365 nm, LED) for Treatment 1 to Treatment 8. The color gradient green–yellow–red represents increasing UV light intensity. Units: mW/cm$^2$.

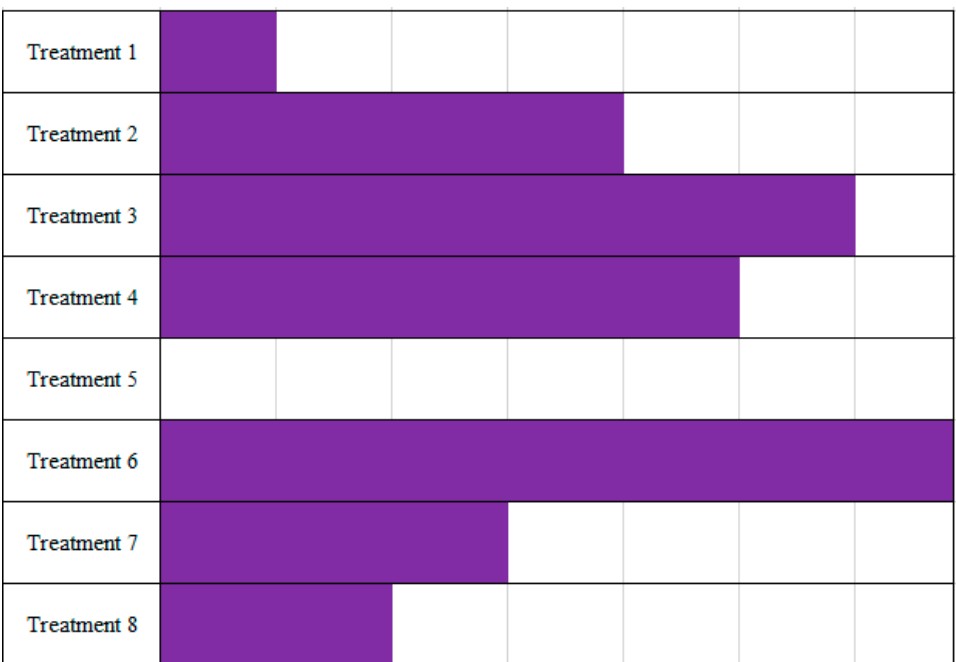

**Figure A9.** The layout of UV irradiation for all eight treatments (top view). Each treatment represents a quartz tube, which is equally "divided" into seven sections. The purple-colored grid indicates UV irradiation areas, while white-colored spaces are shielded from UV. The pictures of this experimental setup with sections of PVC tubes shielding the quartz tubes are provided in Supplementary Materials. The arrangement of the shielding layout was randomly selected and remained the same for the experiments.

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
