# Peer review of "Mitigation of Airborne PRRSV Transmission with UV Light Treatment: Proof-of-Concept"

_agriculture, doi:10.3390/agriculture11030259_

Round 1
Reviewer 1 Report
Dear authors,
In abstractline
21 and other times in the manuscript: mJ/cm2, I think you mean Joule, thus the correct form is MJ, please remember for all the senetences.
line 27. For the first time, I suggest a full description of the acronym (High Efficiency Particulate Air = HEPA)
In introduction
line 33. For example, the acronym U.S. is obvious, no description is neededline 57. Please improve the sentence (in theory...)
line 64. Reported by [20], I mean you can write “reported by Cutler et al, 2012 [20]”
line 113. I checked the bibliographyis ok, however I suggest a better description of MARC-145 cell-line and MA-104.
In materials and methods
line 231. You correctly reported the software, “R studio”. I’m sure that this software has a specific package for your assessment. Can you describe it?In the discussion
Line 332. Prior to the current researchCutler et al. (2012) [20] was the benchmark on UVirradiation on aerosolized PRRSV. Please improve the sentence (grammar). Moreover I saw a similar sentence (line 60), this is a redundance.
Lines 333-335. Other research experiments focused on stationary objects to inactivate PRRSV, e.g., on tissue culture plates [21], on common farm surfaces(rubber, concrete, paper, etc.) [40], on samples inside irradiation chamber [41]. This paragraph is unclear.
Line 340. You adopted the word microbes, although is commonly accepted, I suggest a better one, more suitable for a scientific article (bacteria)
Line 374. I saw, you put the sentence in quotes, however kill the pathogen is a bit reductive for a scientific article
I hope useful, suggestions.
All the best

Reviewer 2 Report
PRRSV is one of the most dangerous diseases that causes a huge economic loss in pig production, thus, prevention and treatment of this disease have been increased the concerns. The paper "Mitigation of Airborne PRRSV Transmission with UV Light 2 Treatment: Proof-of-concept" studied a novel method to control the transmission of this virus with UV light 2 treatment. The manuscript was well written and organized with interesting results. However, there are optional question that I wish get the responses from authors and they do not influence on my decision for current manuscript.
- The effectiveness of UV light treatment is evidenced but the cost of implementing were expensive ($107,600 and $88,440per year per 1000 head swine barn for UV light with pre-filters and HEPA filters, respectively). Do you think these methods can be applied for practical condition in animal farms?
- Do you have any concern that UV light treatment could be harmful to human and animal health?
Thank you
Reviewer 3 Report
General comments:
In this paper, Li et al. assessed and compared different ultraviolet (UV) treatments to inactivate aerosolized PRRSV. Four lamps were tested: UV-A (365 nm, both fluorescent and LED-based), “excimer” UV-C (222 nm), and germicidal UV-C (254 nm). Interesting results were obtained using UV-C lamp, especially “excimer” UV-C. Authors conclude pilot-scale or farm-scale testing of UV-C on PRRSV on PRRSV aerosols simulating barn ventilation rates are recommended based on the effectiveness they observed and the reasonable cost of UV-C lamp comparable to HEPA filtration. PRRSV is virus of major importance in pig farms and biosecurity is often the key in the fight against the virus and associated diseases.
Strengths: Pleasant and well-written. Original research and convincing results.
Weakness: References are not always exactly where they should be. Discussion/comparison with UV-B could be interesting too. Then, a comparison, at least in the discussion, with a naked virus (picornavirus like FMDV for instance), could enrich the discussion.
Major
/
Moderate
-L57: Please add a reference to that sentence.
-L60: Please add a reference after inactivation.
-L65: Please provide doses.
-L67: Harmful effects such as… Please develop.
-L70: Please add a reference after 254 nm.
-L122: Please specify the role of Rhodamine B powder.
-L236-238: Does it make sense to assume two types? What could be the two types for PRRSV-2 here? It might be of interest to mix two types of viruses (naked versus enveloped) to test this two-stage inactivation model.
-Table 3: Higher cost of UV-C 222 nm lamp when the breeder buys it and when he uses it.
-In Table 4: Please specify the type of UV light lamp (C 222 nm I suppose).
-L328: Please add a reference.
-L330: Please add a reference.
-L382: Please be more explicit for the cost and compare.
-Please add the following article in the discussion: DOI: 10.1586/erv.12.38
-Figure B5 is not really clear.
Minor
-L27: Please provide the explanation of HEPA abbreviation (High-efficiency particulate air) before using the abbreviation.
-Some discrepancies in journal name styles, see L478 (full name) and 481 (abbreviated)
Reviewer 4 Report
The article “Mitigation of Airborne PRRSV Transmission with UV Light Treatment: Proof-of-concept” prepared by authors is important, interesting and has merit. Introduction is written well, results are presented clearly however the discussion require extension. This paragraph is mainly repetition of the results. Article need more advanced discussion enrich in other scientific papers. For minor errors I notice:
Line 33: “US” you need write full name first and shortcut later.
Line 40: there is a shortcut “PRRSV” without full name, please write “PRRS virus (PRRSV)”
Line 48: you use shortcut “UV” and in line 56 you use full name and shortcut “Ultraviolet (UV)”, it should be opposite.
Line 63: “Under dynamic conditions (i.e. irradiation on PRRSV aerosols), only a one-stage model was reported by [20].” Reported by who? Write the name of scientist not only a number.
Line 140: ”0.3 x 0.3 m; 1 x 1 ft” – be consequent and use metric!
Line 189: “100.1%” please explain.
Line 216-218: was added FBS or similar serum do the culture medium (if yes in which percent if not explain why)?
Sincerely,
Reviewer
Round 2
Reviewer 4 Report
Authors have answered to all my questions.